# CropHarvest: a global satellite dataset for crop type classification

**Gabriel Tseng**
NASA Harvest
gabrieltseng95@gmail.com

**Ivan Zvonkov**
University of Maryland, College Park
izvonkov@umd.edu

**Catherine Nakalembe**
University of Maryland, College Park
cnakalem@umd.edu

**Hannah Kerner**
University of Maryland, College Park
hkerner@umd.edu

## Abstract

Remote sensing datasets pose a number of interesting challenges to machine learning researchers and practitioners, from domain shift (spatially, semantically and temporally) to highly imbalanced labels. In addition, the outputs of models trained on remote sensing datasets can contribute to positive societal impacts, for example in food security and climate change. However, there are many barriers that limit the accessibility of satellite data to the machine learning community, including a lack of large labeled datasets as well as an understanding of the range of satellite products available, how these products should be processed, and how to manage multi-dimensional geospatial data. To lower these barriers and facilitate the use of satellite datasets by the machine learning community, we present CropHarvest—a satellite dataset of more than 90,000 geographically-diverse samples with agricultural class labels. The data and accompanying python package are available at https://github.com/nasaharvest/cropharvest.

## 1 Introduction

Satellite Earth observation (EO) datasets are rapidly gaining interest in the AI community due to the massive datasets involved as well as the opportunities for addressing urgent challenges related to climate change [42, 49, 46], the environment, agriculture and food security [26, 61, 25, 59], and humanitarian needs [32]. ML systems combined with EO data can be used to develop critical datasets that provide targeted, geography-specific information including early warning of droughts [4, 37, 6], crop failure [47], and pests [11, 9]; impact assessment of climate-related disasters such as flooding and landslides [10]; and tracking and responding to population displacement [62] or health emergencies [44]. This information can lead to life-saving decisions, policies, and emergency operations [56, 3]. However, there are currently many challenges for developing ML systems that use EO data, namely, limited public labeled datasets, a lack of harmonization across labels and source data, and substantial effort required to make EO data "ML-ready" [49, 28, 32].

A major difference from typical ML datasets is that the definition of labels is often decoupled from the model's input data, meaning label annotations are commonly defined in terms of geographic location coordinates (e.g., latitude/longitude) rather than a specific image or image pixels. Collecting these labels can require ground truth observation, which requires substantial time, effort, and cost, and the resulting datasets are rarely made public or consistent [32]. In addition, labels tend to be very imbalanced. Geographic imbalance occurs because some regions have an abundance of large, densely-annotated datasets related to agricultural practices (e.g., in Europe, France's Graphic Plot Register [1] provides a dataset of field boundaries with crop type labels for the whole country and its territories)

35th Conference on Neural Information Processing Systems (NeurIPS 2021) Track on Datasets and Benchmarks.

while datasets in other regions are sparsely-labeled and cover small areas (e.g., in Sub-Saharan Africa, Kehs et al., 2021 provide a dataset of field boundaries with crop type labels for 474 fields in northern Busia county, Kenya) [24]. Semantic imbalance occurs because organizations collect label definitions and metadata depending on their interests, without planning for interoperability. Some efforts collect comprehensive field-level data while others collect very little (if any) information about agricultural practices [8]. The heterogeneity and geographic bias of existing labeled datasets makes it challenging to develop global models and products, and to apply techniques like transfer learning or multi-task learning that could be useful in data-sparse regions. These challenges result in disparate performance of models across geographies as well as difficulties in comparing algorithmic advances in machine learning for geospatial data [51].

These challenges create a high barrier to entry for ML researchers hoping to develop approaches using geospatial data. To address these challenges, we present CropHarvest, a spatially and semantically comprehensive dataset of agricultural class labels. We chose to focus on the task of crop classification for this dataset because it is relevant for many use cases, including food security [35] and climate change assessments [49]. This dataset harmonizes 20 datasets with agricultural class labels, including existing public datasets and new datasets released with this paper. We provide the label geometries as well as the corresponding satellite data inputs from four satellite datasets. We also provide an ML-friendly API inspired by the torchvision package[30] for accessing the dataset to reduce the overhead required for ML practitioners to get started using it. We additionally provide three benchmark tasks for evaluating the performance of models in a range of agroecologies and dataset-size regimes. This dataset can be used to address many challenges of interest to the machine learning community, namely: i) learning from sparse spatio-temporal data, where the spatial correlations are much weaker than in a dense map, ii) handling domain shift in space (regional differences in agriculture and climate), time (inter-annual variability and climate change), and semantics (new crop types or practices), and iii) learning from highly imbalanced datasets, where a few spatially concentrated classes dominate the dataset.

We hope that this benchmark dataset for satellite data will spur new directions of research into ML methods for satellite and geospatial datasets and provide a starting place for AI researchers looking to use satellite data to contribute to global challenges related to climate change and sustainable development. By including datasets and tasks for data-sparse regions, we hope to enable the development of crop type classification models and other ML systems with improved performance in developing regions that are typically under-represented in ML datasets. By evaluating models in a range of geographies, this dataset will also provide a capability for evaluating fairness of ML systems across geographies, which is not possible with existing public datasets.

## 2   Related Work

Remote sensing data have been used with machine learning methods for a wide variety of applications including land cover and land use mapping [65, 57, 52], crop type classification [60, 51], yield forecasting [], flood detection [43], and fuel moisture measurement [45]. While some public global land cover classification datasets exist (e.g., [20, 14, 12]), they often use a limited number of pre-defined land cover labels [68, 58, 34]. The classes included in global land cover datasets are typically chosen to be broadly-applicable and thus do not capture information specific to a particular region (for instance, specific types of crops grown regionally, such as teff in Ethiopia). In addition, the labels are tightly coupled to the satellite data but the original label geometries and pre-processing steps are not available. The choice of satellite data to pair with labels—which can vary by sensor type, spatial resolution, temporal resolution, spectral information, cloud masking or other pre-processing algorithms, and more—greatly influences the ML model outcomes. Not sharing the label geometries and/or pre-processing steps makes it difficult to apply models trained on these datasets to other regions or tasks.

Models that use remote sensing datasets for crop classification often perform better in data-rich regions [50, 54, 38, 33]. The geographic imbalance of available datasets means that models trained using these datasets may not generalize well to other regions where training labels are sparse or non-existent, which are often developing or resource-poor regions. Objects and infrastructure tend to be much smaller and more heterogeneous in developing countries compared to developed countries [36]. These geographic disparities often translate to semantic disparities, as datasets in data-rich regions often contain more agricultural classes and metadata compared to data-sparse regions. This

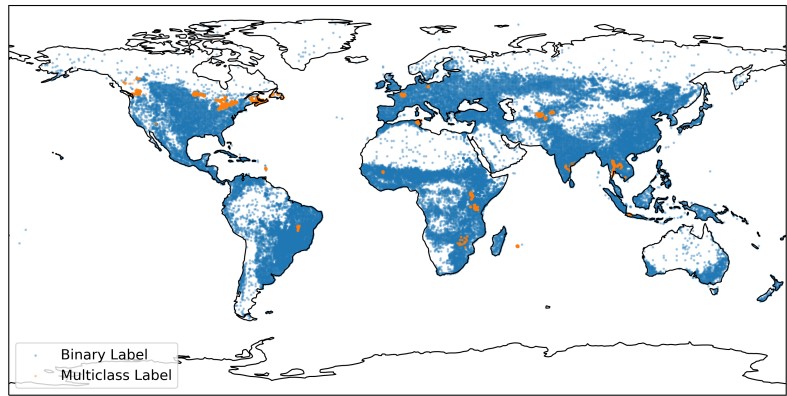

Figure 1: A global collection of crop/non-crop and agricultural class labels, collected from publicly available datasets and covering 343 labels.

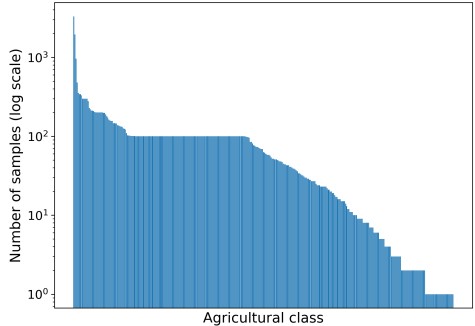

(a) A long tail of agricultural class labels in the dataset (with a log scale on the y-axis), where 55% of classes have less than 100 labels, and 46% have less than 50 labels.

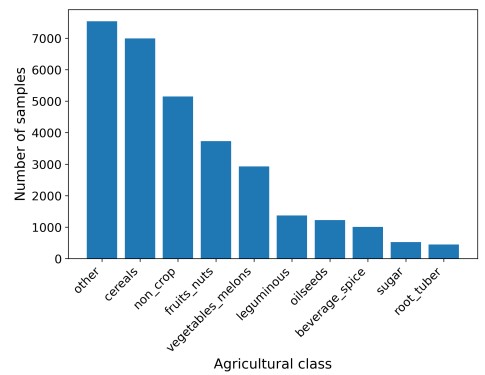

(b) The distribution of [13] crop category labels in the dataset.

Figure 2: The distribution of labels in the dataset for (a) raw labels and (b) [13] crop category labels.

in turn hinders the capability of models trained on datasets from data-rich regions to transfer well to new areas. There is an increased recognition from the agricultural monitoring community (including the public and private sector) that satellite-driven ML systems hold significant potential to address critical agricultural information gaps. Addressing these limitations in existing datasets is particularly important for developing ML solutions for developing countries and regions dominated by smallholder farming that lack readily accessible ground-truth data [40] and where there is an urgent need for ML solutions that help address food security and climate change identified by a wide range of end-users across the agricultural sector [6].

## 3 The CropHarvest dataset

The CropHarvest dataset is a crop dataset of geo-referenced labels with satellite data inputs, each consisting of latitude, longitude, the associated agricultural label, and a satellite pixel time series. In the following sections, we discuss how the labels were collected (section 3.1) and the satellite data products used to create the inputs associated with each label (section 3.2). Code to access the dataset is available at `https://github.com/nasaharvest/cropharvest` and the raw data is available at `https://zenodo.org/record/5533193`. The dataset is released under a CC BY-SA 4.0 license.

### 3.1 Labeled Data

We collected 90,480 datapoints from 20 datasets; some datasets come from existing public sources while some (e.g., Rwanda) are being made public with this publication (Table 1). All of the datapoints

Table 1: A list of datasets combined, including their source, country of focus, the number of labels used in `CropHarvest` and the license. Datasets with a binary label type contained binary "crop" or "non crop" labels. The first group of rows includes datasets from NASA Harvest, the second from [22], and the third from a variety of other sources. Datasets listed in order of size within each group.

| Source | Area of Focus | # Labels | Label Type | License |
|---|---|---|---|---|
| NASA Harvest | Rwanda | 3,600 | Binary | |
| | Kenya | 2,704 | Binary | |
| | Ethiopia | 830 | Binary | |
| | Sudan | 422 | Binary | CC BY-4.0 |
| | Mali | 142 | Binary | |
| | Brazil | 36 | Binary | |
| [25] | Togo | 1,582 | Binary | |
| [22] | France (Ile de France) | 6,184 | Multi-class | |
| | France (Réunion) | 2,776 | Multi-class | etalab Open License |
| | France (Martinique) | 2,421 | Multi-class | |
| [55] | Global | 35,866 | Binary | CC BY-3.0 |
| [66] | Global | 14,976 | Multi-class | LP DAAC[1] |
| [2] | Canada | 9,088 | Multi-class | Open Government License |
| [48] | Uzbekistan, Tajikistan | 5,302 | Multi-class | CC BY-4.0 |
| [27] | Germany | 2,550 | Multi-class | DL-DE->BY-2.0 |
| [39] | Brazil | 800 | Multi-class | CC BY-4.0 |
| [19] | Tanzania | 392 | Multi-class | CC BY-4.0 |
| [23] | Kenya | 319 | Multi-class | CC BY-SA-4.0 |
| [5] | Uganda | 233 | Multi-class | CC BY-4.0 |
| Harvest Partner | Mali | 148 | Multi-class | CC BY-4.0 |
| FEWS NET | Zimbabwe | 49 | Multi-class | CC BY-SA-4.0 |

[1] All LP DAAC current data and products acquired through the LP DAAC have no restrictions on reuse, sale, or redistribution.

have a binary label specifying crop or non-crop. Of the 90,480 total datapoints, 30,899 (34.2%) have more granular labels spanning 348 labels. We refer to these labels as "agricultural class labels" because they consist mostly of crop types (e.g., "maize") but also contain additional agriculture-related classes (e.g., "pasture"). Figure 1 shows the spatial distribution of these datapoints with emphasis on the locations of datapoints with granular labels. Figure 2a shows the number of examples in each class, illustrating the high class imbalance in the dataset. For these 90,480 datapoints with granular labels, we included an additional higher-level category label. We determined these classes using the Food and Agriculture Organization (FAO)'s indicative crop classifications [13]. These consist of 9 crop type groupings: cereals, vegetables and melons, fruits and nuts, oilseed crops, root/tuber crops, beverage and spice crops, leguminous crops, sugar crops, and other crops. We grouped all non-crop classes (e.g., water or forest) into a tenth non-crop class. Figure 2b shows the distribution of these categories in the dataset. Similar to the granular agricultural class labels, these crop categories also have substantial class imbalance.

We applied some filters to the original source datasets. We only included labels collected after 2016 to ensure satellite data could be acquired for each label; at the time of export, Google Earth Engine Sentinel-2 L1C data was available starting from June 2015. We excluded labels that contained multiple crop classes and/or were annotated as inter-cropped (i.e., multiple crops planted in one field). Finally, we undersampled datasets that contained substantially more samples than other datasets so they would not dominate the combined dataset; in the current version, these were the [22] and [2] datasets. We sampled at most 100 labels from each crop type in each of these datasets.

## 3.2 Remote sensing data

Each label has either a polygon (24% of data points) or point (76% of data points) geometry, depending on the method used to collect the labels. We paired each label geometry with the corresponding satellite data. Satellite observations are typically in raster format, which is a grid of pixels where

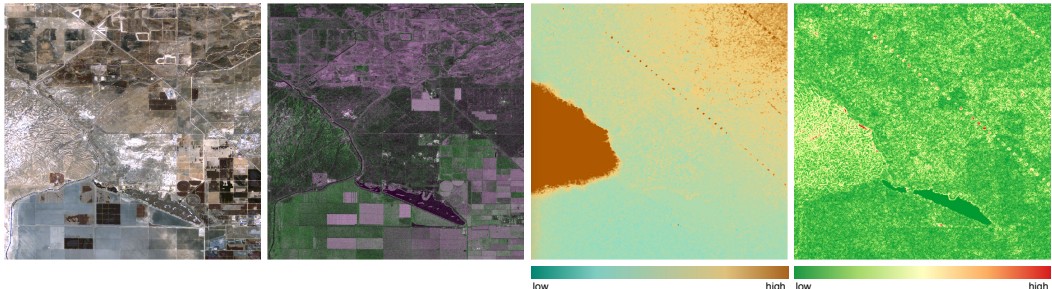

Figure 3: Satellite data for a sample region in southern California, USA. From left to right: Sentinel-2 RGB image, Sentinel-1 image (false color: VH-VV-VH), elevation, and slope. The Sentinel-2 RGB and Sentinel-1 SAR images are from the 5th time step in the series (June) and the elevation and slope images are constant for the whole time series.

each pixel contains a value representing the observed information at a specific latitude/longitude location, such as temperature or reflectance. For each point label, the pixel with location nearest to the label location is selected. For each polygon label, the pixel nearest to the centroid of the polygon is selected to prevent oversampling of polygons compared to points. For each label, a one-year time series with 12 timesteps is constructed, each representing an aggregated value over 30 days.

We included satellite data from four datasets that provide useful and complementary information for discriminating agricultural classes. All datasets are publicly available. We used Google Earth Engine to access the data and upsample all products to the same spatial resolution of 10 m/px. We have made all code used to acquire the satellite data open source to enable machine learning practitioners to generate crop classification maps in new areas and more easily acquire satellite datasets for other tasks.

**Multispectral Optical Images** Multispectral information is critical for crop identification: a crop's composition, growth stage, canopy structure, and leaf water content can all affect how it reflects light at different wavelengths [67]. We used Sentinel-2 multispectral observations since this dataset has the highest spatial (10-60 m/px) and temporal (5 day revisit) resolution of current publicly-available satellite datasets. Sentinel-2 has 13 spectral bands including the visible color RGB wavelengths typically used in ML datasets, near-infrared wavelengths which are useful for detecting chlorophyll, and short wave infrared (SWIR) wavelengths which are sensitive to water content [7]. We used the Sentinel-2 Top of Atmosphere Reflectance (Level 1C) available on Google Earth Engine [link]. Optical satellite images contain cloud artifacts that need to be removed. We constructed a cloud-free time series of Sentinel-2 images by using [53] to find the least-cloudy pixel within each 30-day window. We used all bands except B1 (coastal aerosols, used to detect fine particles in the air or contaminants in the water) and B10 (cirrus SWIR, used for cloud detection). In addition, we appended normalized difference vegetation index (NDVI) which is the normalized difference between the near-infrared (B08) and red (B04) bands ($NDVI = \frac{B08 - B04}{B08 + B04}$). Because vegetation absorbs red light and reflects near-infrared light, high NDVI values often indicate healthy vegetation [63].

**Synthetic Aperture Radar (SAR) Data** SAR differs from optical imagery in that instead of passively measuring light reflected from the Earth, SAR sensors beam down radio signals and measure what is reflected back, providing information about about the geometry and water content of the crop. SAR sensors can penetrate cloud cover, making it useful for providing coverage in very cloudy regions and seasons. We used the Sentinel-1 C-band Synthetic Aperture Radar (SAR) Ground Range Detected (GRD) dataset [link]. Sentinel-1 has a 10m resolution with near-global coverage and a highly variable revisit time depending on the region of interest (ranging from several days to several months). For each input, we used either imagery taken during an ascending or descending orbit (depending on which was available for the location). Sentinel-1 emits and receives radio signals at certain polarizations. We used the VV (emit at a vertical and receive at a vertical polarization) and VH (emit at a vertical and receive at a horizontal polarization) bands. We took the median of all available observations within each 30-day window. If no observation was available in the window, we used the temporally closest available observation for that location.

Table 2: The distribution of pixels labelled as positive and negative examples in the benchmark evaluation data.

| Task | Total | Positive (%) | Negative (%) |
|---|---|---|---|
| Kenya | 898 | 575 (64.0%) | 323 (36.0%) |
| Brazil | 682,559 | 174,026 (25.5%) | 508,533 (74.5%) |
| Togo | 306 | 106 (34.6%) | 200 (65.4%) |

**Meteorological Data**    Crops have distinct spectral-temporal profiles and crop development can be delayed or accelerated by weather conditions [64, 16]. Variations in climate should be considered when classifying crop types, particularly the temperature and precipitation distribution throughout the growing seasons [16]. We used the ERA5 meteorological reanalysis dataset [link], which provides a variety of meteorological data globally at 31 km/px and hourly resolution to capture this information. While this product has substantially coarser resolution than the other datasets, it could still be useful for enabling models to learn regional context. We used the monthly means product, which gives the monthly average of the reanalysis dataset. We included total precipitation and ground temperature (at 2 m height) from the ERA5 dataset. For each input, we selected the month with the most overlap with each 30 day time period.

**Topographic Data**    The topography of an area can affect its suitability for certain crops [31]. The Shuttle Radar Topography Mission (SRTM) Digital Elevation Model (DEM) [link] provides global elevation at 30 m/px resolution. For each label, we selected the elevation of the latitude and longitude nearest to the label location. We used the surrounding elevations to calculate the slope.

### 3.3  Limitations

While we made an effort to address as many limitations as possible in the creation of the dataset, some limitations remain. The agricultural classes in the dataset are not globally harmonized. This is because the original classes do not perfectly overlap, so labels could not be harmonized without coarsening the labels. For example, the [23] dataset contains a "groundnut" class, and the [66] dataset contains a "groundnuts or peanuts" class; combining these classes would require losing the more granular class. Instead, all instances are assigned a "crop" or "non-crop" value, and are assigned a crop category from the indicative crop classification [13]. Another limitation is incomplete satellite time-series coverage for some labels. Due to the limited satellite data coverage (in particular for Sentinel-1 data), satellite data was obtained for 65,690 (73%) of the collected labels.

## 4  Benchmarks

We defined a set of benchmark tasks to enable comparison of machine learning models in a wide range of agroecologies and in-distribution dataset size regimes. To this end, we selected three geographically and semantically distinct tasks with training data sizes ranging from $< 100$ to $> 1000$. When trained models are used to predict dense maps (e.g., to generate a predicted crop type map of a region), there are two commonly used methods for evaluating the resulting map: random sampling and wall-to-wall evaluation. We included both evaluation metrics in the benchmark tasks so that success on these tasks will better translate to success in real world application of the models. Figure 4 illustrates the landscapes covered by each task in high-resolution satellite images.

We include a breakdown of the ratio of pixels labelled as positive and negative pixel in the evaluation data in Table 2. The large pixel count in the Brazil task stems from the use of entire polygons for evaluation, instead of polygon centroids.

### 4.1  Crop type vs. rest

In two of the benchmark tasks, the objective is for a model to classify whether a pixel contains a specific crop type or not, which we refer to as *crop type vs. rest*. We designed two tasks with different agroecologies and crop types: Kenya (maize vs. rest) and Brazil (coffee vs. rest). For both tasks, we undersampled non-crop samples so that the negative samples consisted of an equal number of

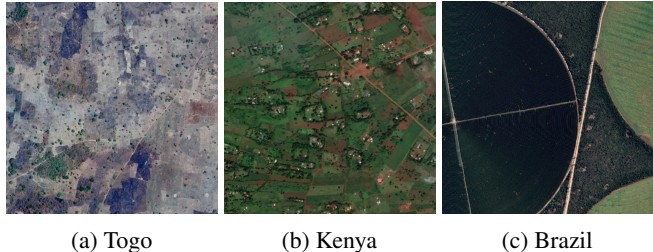

|          |          |           |
|:--------:|:--------:|:---------:|
| (a) Togo | (b) Kenya | (c) Brazil |

Figure 4: Example 1km × 1km satellite images of the evaluation regions, demonstrating the variety in field sizes and agro-ecologies being evaluated. (Images obtained from Google Earth Pro basemaps, comprised primarily of high resolution Maxar images.)

non-crop samples and other-crop samples. This is because it is more difficult to differentiate between crop types (e.g., maize vs. sorghum) than between crop and non-crop instances. The label geometry for the test datasets in both tasks are polygons rather than points. To increase the number and diversity of samples in the test sets, we constructed the test set for each task by defining a bounding box within the region covered by the dataset. We then evaluated the model predictions for all pixels in the polygons contained in the bounding box, rather than only evaluating each polygon centroid. We removed the test polygons from the training sets to ensure there was no spatial overlap between the training and test sets, which could result in spatial autocorrelation [28]. For both tasks, the evaluation bounding boxes were selected because they included a good balance of positive and negative classes.

**Kenya: maize vs. rest**  The goal in this task is to classify samples in Kenya as maize vs. rest using a dataset collected by PlantVillage [23]. This dataset consists of a training set of 1,345 samples (266 maize and 1,079 rest) collected in 2020-2021. We used two evaluation bounding boxes with labels from 2020 which covered approximately $764 \text{ m}^2$ and $1,178 \text{ m}^2$ (total coverage of $1,942 \text{ m}^2$), resulting in 45 polygons total. We used two boxes to increase the diversity of test samples.

**Brazil: coffee vs. rest**  We used the LEM+ dataset [39] to construct a coffee vs. rest task in Brazil. The training set has 794 samples (21 coffee, 773 rest) collected in 2020-2021. The test set included all polygons from 2020 or 2021 within a $4.2 \text{ km}^2$ bounding box, resulting in 66 polygons total.

## 4.2   Crop vs. non-crop (Togo)

We used the dataset collected by [25] to construct a binary crop vs. non-crop classification task in Togo. This dataset contains 1,319 samples in the training set 1,319 and 306 samples in the test set.

In this task, we included an additional experiment to investigate model performance as a function of training data size. We evaluated the models after training on a range of sizes subsampled from the original training dataset, ensuring each subset consisted of an equal number of positive and negative samples: $\{20, 50, 126, 254, 382, 508, 636, 764, 892, 1020, 1148, 1319\}$. Each subsample adds additional samples to the previous (smaller) subsample, allowing us to measure the effect of additional training samples on model performance.

## 4.3   Models

We evaluated the performance of three neural network models and a random forest model on the benchmark tasks. Random forests are commonly used for remote sensing tasks [29], so we included it as a benchmark to compare with the deep learning models. We implemented the random forest using default settings in [41] and trained it on the evaluation tasks. All neural network models have the same architecture, but differ by the method used to initalize their weights: a 1-layer long short-term memory network (LSTM) [21] with a hidden vector size of 128. The final hidden layer outputs are passed to a 2-layer classifier and a sigmoid activation. We applied variational dropout [17] between each LSTM timestep with 20% of weights randomly dropped (dropout value of 0.2). We used three different weight initialization methods for this architecture:

Table 3: Benchmark results. All results are averaged from 10 runs to account for differences in random seeds; results are reported with the accompanying standard error. We report the area under the receiver operating characteristic curve (AUC ROC) and the F1 score using a threshold of 0.5 to classify a prediction as the positive or negative class. The best metric for each task is in **bold**.

| Task | Model | AUC ROC | F1 |
|------|-------|---------|-----|
| Kenya | Random Forest | $0.578 \pm 0.006$ | $0.559 \pm 0.003$ |
| | Random | $0.329 \pm 0.011$ | $0.782 \pm 0.000$ |
| | Pre-trained | $0.694 \pm 0.001$ | $0.819 \pm 0.001$ |
| | MAML | $\mathbf{0.729 \pm 0.001}$ | $\mathbf{0.828 \pm 0.000}$ |
| Brazil | Random Forest | $\mathbf{0.941 \pm 0.004}$ | $0.000 \pm 0.000$ |
| | Random | $0.898 \pm 0.010$ | $\mathbf{0.764 \pm 0.012}$ |
| | Pre-trained | $0.820 \pm 0.002$ | $0.619 \pm 0.005$ |
| | MAML | $0.831 \pm 0.005$ | $0.496 \pm 0.001$ |
| Togo | Random Forest | $0.892 \pm 0.001$ | $\mathbf{0.756 \pm 0.002}$ |
| | Random | $0.861 \pm 0.002$ | $0.720 \pm 0.005$ |
| | Pre-trained | $\mathbf{0.894 \pm 0.000}$ | $0.713 \pm 0.002$ |
| | MAML | $0.878 \pm 0.001$ | $0.662 \pm 0.001$ |

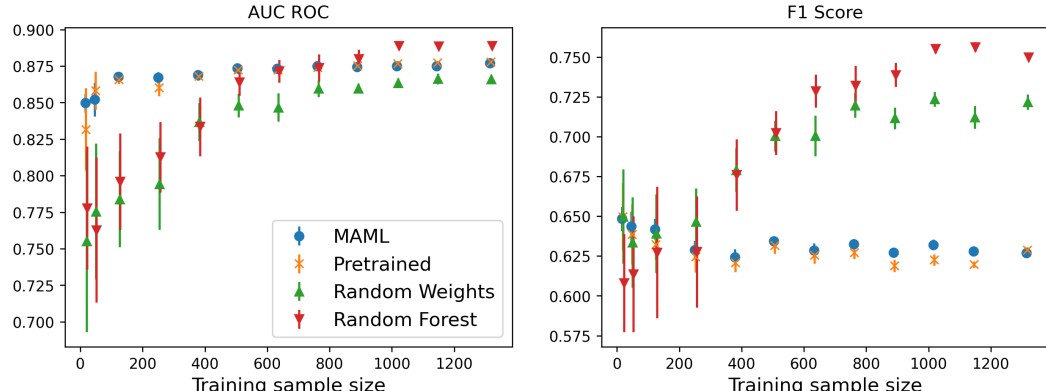

Figure 5: Benchmark results for the Togo test task as a function of dataset size. All results are averaged from 10 runs; results are reported with error bars representing the standard error. We report the area under the receiver operating characteristic curve (AUC ROC) and the F1 score.

- **Model Agnostic Meta Learning (MAML)** We initialized the neural network weights using MAML [15]. We used the entire dataset for training, excluding the test tasks. We defined tasks as in [60]: we used country bounding boxes to spatially partition tasks and defined crop vs. non-crop and crop type vs. rest tasks. We then finetuned the MAML model on the evaluation tasks.

- **Crop vs. non-crop pre-training** We pre-trained the neural network to predict crop vs. non-crop for the entire dataset. We then fine-tuned the model on the evaluation tasks.

- **Random weights** We initialized the neural network with random weights as described in [18] then trained it on the evaluation tasks.

We trained all models on the task of interest for 250 gradient steps. We used a batch size of 20, where each batch contained 10 positive and 10 negative instances. We used stochastic gradient descent to optimize model weights. We ran all experiments on an AWS t2.xlarge instance. The code to run the benchmarks is at `https://github.com/nasaharvest/cropharvest`.

## 4.4 Results

The results for the four models on each benchmark task are reported in Table 3. The Togo task results are for the models trained on the entire training dataset. Figure 5 shows the performance

of the models for the Togo task as a function of training data size, as described in Section 4.2. No single model performs best across all benchmark tasks, which illustrates the different challenges and scope for future research captured by this this dataset and the proposed benchmarks. Random forests perform well across all tasks, but the poor F1 scores in Kenya and Brazil suggest they may be poorly calibrated for imbalanced tasks. For the deep learning models, first training on the entire dataset (using pre-training or MAML) has clear benefits and improves AUC ROC relative to the randomly initialized model for all benchmark tasks. This suggests the global dataset can help the model learn useful representations for local tasks. This benefit is especially notable for the Kenya benchmark and for the Togo task with smaller training sample sizes. However, pre-training and MAML result in very different performance across the tasks, suggesting that the same weight initialization strategy may not give the best performance for different datasets, even if they are similar applications or data types.

## 5  The `cropharvest` package

A key aim of this project is to enable machine learning and remote sensing practitioners to 1) use this dataset as a starting point for working with satellite data and sustainable development challenges, 2) use this dataset to train new models and improve performance on the benchmark tasks, 3) generate classification maps for new areas, especially those underrepresented in existing remote sensing derived products, and 4) contribute their own data to this dataset. We hope that creating a coherent global dataset will allow models to be trained and evaluated for data-sparse regions and improve the geographic diversity and fairness of machine learning and geospatial systems. To facilitate data access, we provide a curated code base for access to the labels, satellite data and processed files, and the models trained for the benchmark tasks at `https://github.com/nasaharvest/cropharvest`. This repository also includes instructions for how to contribute new datasets to CropHarvest. Below is a minimal working example drawn from the `demo.ipynb` notebook in the GitHub repository. The code in this example downloads the labels GeoJSON and the associated HDF5 satellite data arrays to train a random forest model for the Togo evaluation task. Additional examples are also available in the GitHub repository.

```python
from cropharvests.datasets import CropHarvest
from sklearn.ensemble import RandomForestClassifier

evaluation_datasets = CropHarvest.create_benchmark_datasets("data")

# evaluation_dataset is a list of evaluation datasets,
# where the Togo dataset is the final list element
togo_dataset = evaluation_datasets[-1]
X, y = togo_dataset.as_array(flatten_x=True)

model = RandomForestClassifier()
model.fit(flatten(X), y)

test_preds, test_y = [], []
for _, test_instance in togo_dataset.test_data(flatten_x=True):
    test_preds.append(model.predict_proba(test_instance.x)[:, 1])
    test_y.append(test_instance.y)
```

### 5.1  Extending CropHarvest and versioning

For CropHarvest to remain up to date as new agricultural datasets are released, we plan to incorporate new datasets as they become available and allow others to do the same. (As new datasets are added, new benchmark tasks may also be added but existing task definitions will not be changed to maintain inter-comparison.) To this end, we decoupled the input datasets from the CropHarvest package so that only a single file needs to be added for a new dataset to be added to CropHarvest. Instructions for how to do this are in the GitHub repository. One challenge resulting from this decoupling is that direct algorithmic comparisons using CropHarvest depend on the data being consistent. We therefore couple the CropHarvest python package version with the constituent datasets to ensure that models trained using the same version of CropHarvest will be trained using the same data. This enables the dataset to be extended in the future while ensuring algorithms can still be compared.

# 6 Conclusion

We present CropHarvest, a global crop dataset of over 90,000 samples assembled from a variety of labelled datasets with satellite data inputs from four remote sensing data products. This dataset is intended to spur new directions of ML research for satellite and geospatial datasets and provide a starting place for AI researchers looking to use satellite data to contribute to global challenges. By including labels and tasks in data-sparse regions (e.g., Sub-Saharan Africa), we hope to enable the development of improved models in developing regions that are typically under-represented in ML datasets. This dataset will also enable the research into methods that incorporate geographic or spatial information, since the location metadata is provided in addition to the input data and labels. In addition to the CropHarvest dataset, we proposed three geographically and semantically diverse evaluation tasks which test models in a range of agroecologies. By evaluating models in a range of geographies, this dataset will also provide a capability for evaluating fairness of ML systems across geographies, which is not possible with existing public datasets. We aim to facilitate use of CropHarvest by releasing the data with a python package which mimics the torchvision API to enable machine learning practitioners to easily begin interacting and developing models with the dataset.

## Acknowledgments

We thank Radiant Earth Foundation for publishing several datasets included in CropHarvest. Zimbabwe data collection and archiving was supported by the Famine Early Warning Systems Network and the Zimbabwe Ministry of Agriculture. The Mali dataset was collected in collaboration with NASA Harvest Partner Lutheran World Relief. We thank Dr. Inbal Becker-Reshef for reviewing and providing valuable feedback. This work contains information licensed under the Open Government Licence – Canada. This work was supported by the NASA Harvest Consortium (Award #80NSSC17K0625), Helmets Labeling Crops grant from the Lacuna Fund (Award #305334-00001), Earth Observations for Field Level Agricultural Resource Mapping (EO-FARM) grant from the SwissRe Foundation (Award #302916-00001), Estimating Cropped Area and Production in the Feed the Future/Mali Zone of Influence grant from NASA Goddard Space Flight Center and USAID (Award #3302915-00001), and Earth Observation for National Agricultural Monitoring grant from NASA SERVIR (#80NSSC20K0264).

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
