# OpenReview forum: "CropHarvest: A global dataset for crop-type classification"
_NeurIPS.cc/2021/Track/Datasets_and_Benchmarks/Round2 — NeurIPS 2021 Datasets and Benchmarks Track (Round 2)_

### Official Review · Reviewer_GTFT · 2021-09-19
**a interesting global satellite dataset for crop type classification**

**Rating:** 7
**Confidence:** 4
**Correctness:** yes
**Clarity:** yes

**Strengths:**

The author makes reasonable use of existing satellite products to obtain the large dataset.
The experimental analysis is reasonable

**Weaknesses:**

The format of the article needs to be improved

**Additional Feedback:**

no

**Documentation:**

yes

**Relation To Prior Work:**

yes

**Summary And Contributions:**

The author presents the CropHarvest—a satellite dataset of nearly 90,000 geographically-diverse samples with agricultural labels.
The author makes reasonable use of existing satellite products to obtain the large dataset.

---

> ### Author Response · Authors · 2021-09-23
> **Response to Reviewer #3**
>
> Thank you for your review. In response to the comment about the format of the article, we have updated the paper in response to comments from reviewer 1, with the goal of improving the clarity of the class descriptions. If this does not address your suggestion, can you please be more specific about how you suggest we improve the format?

---

### Official Review · Reviewer_kgNg · 2021-09-19
**CropHarvest**

**Rating:** 8
**Confidence:** 4
**Clarity:** The paper is well written.

**Strengths:**

+ Data richness.
+ Broad interest for both, pure ML researchers and also for ML researchers interested in climate change and sustainable development.
+ Global coverage (world level).
+ Accessibility.
+ It will also enable the research into methods that incorporate geographic or spatial information since the location metadata is provided in addition to the input data and labels.

**Weaknesses:**

+ There are some limitations of the dataset related to the representativeness of the data, but this is a minor issue once it has been identified.

**Additional Feedback:**

Good work!

**Correctness:**

The dataset is constructed in a sound way and a large effort has been devoted to lowering the existing barriers to access this kind of data.
The evaluation of the benchmark task is well done and the conclusions are very informative.

**Documentation:**

+ Data collection and organization are documented.
+ There is a plan to ensure the availability and maintenance of the dataset.
+ CropHarvest has a Creative Commons Attribution-ShareAlike 4.0 International license.
+ CropHarvest Datasheet is available.

**Ethics:**

There are no specific ethical concerns.

**Relation To Prior Work:**

Existing work has been extensively reported in the paper.

**Summary And Contributions:**

+ CropHarvest is a satellite dataset of nearly 90,000 geographically-divers samples with agricultural labels. This dataset harmonizes 20 datasets with crop type labels, including existing public datasets and new datasets released with the paper.
+ An ML-friendly API inspired by the torchvision package is released for accessing the dataset.
+ There are three benchmark tasks for evaluating the performance of models in a range of agro-ecologies and dataset-size regimes.
+ Data is very rich. There are Multispectral Optical Images, Synthetic  Aperture  Radar  (SAR)  Data, Meteorological Data, and Topographic Data.

---

> ### Author Response · Authors · 2021-09-23
> **Response to Reviewer #2**
>
> Thank you for your review and for the comments. We respond to them below:
>
> #### 1. Representativeness of the data
> We tried to ensure the dataset was globally representative, but recognize that - especially for data points with more granular labels - certain regions remain under-represented. We hope that future iterations of the dataset will improve representation in under-sampled regions as we and external contributors add new and existing datasets.

---

> > ### Comment · Reviewer_kgNg · 2021-09-28
> > **CropHarvest (Final rating)**
> >
> > I definitely think that this is a good paper about a very interesting dataset. Kudos!

---

### Official Review · Reviewer_hf2F · 2021-09-21
**CropHarvest unifies a variety of previously fractured public and private datasets to accelerate machine-learning based applications of global agricultural production.**

**Rating:** 8
**Confidence:** 4
**Clarity:** The paper is clearly written and easy…

**Strengths:**

As someone who works in the agricultural/remote sensing nexus, I can attest that a lack of harmonized labels and substantial effort to make EO data “ML-ready” is an enormous challenge. It is because of this that only some of the most well-funded groups have been able to pursue analyses in these areas, and many projects have withered on the vine for lack of ability to use such data. This is an important and timely contribution that I suspect will prove very useful for advancing research in this area and be well cited.

I further applaud the team for encouraging additional contributions to the dataset and providing instructions for new contributions to enhance its utility across an even broader set of actors and regions as new data emerges. Bravo!


**Weaknesses:**


Unless I’m missing detail, it seems the authors may be conflating the definition of land use (economic uses of land) with land cover (different types of crops). It seems that perhaps land cover may be a closer fit. This might seem like a minor issue, and many times land use and land cover appear together (like in the NASA LCLUC program of which the authors I’m sure are connected to), but several people get rather itchy around the specific definition of each of those terms, so at least I’d recommend the authors review which they intend.

Although I understand the desire to include meteorological data as part of the product options, the 31km spatial resolution seem much too coarse a resolution be useful for most agricultural applications. However, I am open to being wrong on this and will be curious to see what other researchers do with the data at larger scales. The other attached remotely sensed datasets (sentinel-1 SAR data, least cloudy Sentinel-2 images, and the SRTM DEM) are very reasonable.

Also it’s an important first step to have land cover labels – an additional hope is that a similar effort can be done for evaluating agricultural yields.


**Additional Feedback:**

I’m looking forward to this paper and the dataset being published so I can share it with others. Kudos on an excellent and important effort. I foresee this being well-cited.

**Correctness:**

The dataset appears to be constructed in a thoughtful way and is well documented.

**Documentation:**

Very clear documentation, availability, and intended use. Clear URL. They have provided instructions for future contributions, although unclear what the future maintenance plan is (to be fair, unclear what the maintenance plan is for *any* ML based dataset I’ve reviewed so far – this one has provided at least some productive outlet for future contributions to consistently help the dataset grow further).


**Ethics:**

None known

**Relation To Prior Work:**

The manuscript reflects a clear linkage to the existing problems in the literature and a knowledge of what their dataset has the potential to enable.

**Summary And Contributions:**

This dataset (CropHarvest) contains nearly 90,000 geographically-diverse samples of the earth’s agricultural lands, labeled and made ready for ML-based applications by linkages with four remote sensing data products. This dataset is spatially and semantically comprehensive as well as coherent. The authors test out four sample different models to evaluate the performance of the model across three sample locations – Togo, Brazil, and Kenya – and find that model performance varies by location but that the global dataset offers some useful performance improvements even for these localized tasks. The dataset offers significant value in reducing the amount of pre-processing needed for spatially explicit ag research in a way that can accelerate research on applications relating to global food security, climate change, and agricultural development.

---

> ### Author Response · Authors · 2021-09-23
> **Response to Reviewer #1**
>
> Thank you for your review. We are very happy to hear you think this will be a useful contribution to the ML and agriculture community.
>
> We appreciate your thoughtful comments and suggestions, and address them below:
>
>
> #### 1. Conflating land use with land cover
> Thank you for pointing this out - we recognize this may be confusing, and have updated the definitions in the paper. We focus on land cover classes, but include land use classes if those are available. We therefore now refer to the classes as “agricultural classes” in the paper to avoid confusion with existing terms.
>
> #### 2. Spatial granularity of the meteorological data
> We agree that ideally, more spatially granular meteorological data would be used and we would be enthusiastic to add such a product in the future if one becomes globally available in the Google Earth Engine data catalog. Even with the coarser resolution, we believe this data can still be useful for models to learn regional context (and some experiments we have run suggest this is the case). We added a sentence to the revised version to clarify this.
>
> #### 3. Evaluating agricultural yields
> We agree that a comprehensive dataset on yields would be beneficial and hope that in the future we and other community members can work towards this goal.

---

### Decision · Program_Chairs · 2021-10-09

**Decision:**

Accept

**Comment:**

All reviewers recommend acceptance and the AC has no reason to overturn the reviewers' decision. What an exciting dataset!! Congratulations to authors!